# Designing Hollow Carbon Sphere with Hierarchal Porous for Na-S Systems with Ultra-Long Cycling Stabilities

**DOI:** 10.3390/molecules27185880

**Published:** 2022-09-10

**Authors:** Gongke Wang, Yumeng Chen, Shaohui Yuan, Peng Ge

**Affiliations:** 1School of Materials Science and Engineering, Henan Engineering Research Center of Design and Recycle for Advanced Electrochemical Energy Storage Materials, Henan Normal University, Xinxiang 453007, China; 2School of Minerals Processing and Bioengineering, Central South University, Changsha 410083, China

**Keywords:** room-temperature Na-S battery, electrode materials, high storage capacity, porous carbon material

## Abstract

Captured by the low-cost and high theoretical specific capacity, Na-S systems have garnered much attention. However, their intermediate products (dissolved polysulfide) are always out of control. Considering the excellent space confinements and conductivity, they have been regarded as promising candidates. Herein, the hollow spheres with suitable thickness shell (~20 nm) are designed as hosting materials, accompanied by in-depth complexing. Benefitting from the abundant micro-pores (mainly about conical-type and slits-type pores < 1.0 nm), the active S4 molecules are successfully filled in the pores through vacuum tube sealing technology, effectively avoiding the process from solid S_8_ to liquid Na_2_S_6_. As cathode for Na-S systems, their capacity could remain at 920 mAh g^−1^ at 0.1 C after 100 cycles. Even at 10.0 C, the capacity still remained at about 310 mAh g^−1^ after 7000 cycles. Supported by the detailed kinetic behaviors, the improvement of ions diffusion behaviors is noted, bringing about the effective thorough redox reactions. Moreover, the enhanced surface-controlling behaviors further induces the evolution of rate properties. Therefore, their stable phase changing is further confirmed through in situ resistances. Thus, the work is anticipated to offer significant design for hosting carbon materials and complexing manners.

## 1. Introduction

Attracting researchers by its high theoretical specific capacity (>1000 mAh g^−1^), metal–sulfur batteries have been of great interest since the 1960s. As a popular type of this, Na-S displays considerable capacity of about 1675 mAh g^−1^ with rich resources. As is well known, the traditional Na-S system belongs to high-temperature systems (>300 °C), accompanying the utilization of solid *β*-alumina (NaAl_11_O_17_) electrolyte, but bringing about the unsafe factors and limited application [1,2,3,4]. Therefore, room-temperature (RT) Na-S systems have captured much attention with the use of liquid electrolytes. Despite the advantages above, it should be not neglected that RT Na-S systems suffer from relative low conductivities, mainly resulted from natural sulfur (5 × 10^−30^ S m^−1^). Compared to the radii of lithium-ions (0.76 Å), the larger radii of Na-ions (1.02 Å) has led to sluggish reactions with S-atoms, accompanying the inconsiderable rate properties [5,6,7]. Moreover, owing to the inferior electronegativity of S-elements (2.5), the slow kinetics of alloying-reactions induced the formation of polysulfide (Na_2_S_n_, 2 ≤ *n* ≤ 8), bringing about deteriorated electrochemical properties, especially shortened cycling life [8,9].

In order to solve the above issues, a series of hosting matrix was designed, such as carbon materials, conductive polymers and metal-oxide/sulfides [10]. Although the shuttling effect of the by-product was hardly controlled, the effective alleviations could induce the evolution of electrochemical properties. Moreover, considering the high cost of the latter two, carbon materials with low cost and great adsorption abilities have been deemed as promising candidates, achieving remarkable progress [10,11]. Regarding the design of carbon materials, the increased contacting area and abundant pore structure were expected [12]. Among them, the simple design of larger specific surface area could bring about the high loading of S materials, but this hardly captures the dissolution of polysulfide [7,13]. Of course, through the tailoring of porous structures, the loading area could be still reached. On the other hand, the limited porous accommodating volume could serve as the restricted site (that is to say, geometry confinement), finally inhibiting the moving of polysulfide [13,14]. It is known that, in the traditional Na-S systems, the cyclooctasulfur (cyclo-S_8_) usually existed with multi-stepwise reactions (from S_8_ to Na_2_S). Moreover, the dissolution of Na_2_S_n_ (4 ≤ *n* ≤ 8) has been confirmed. From the previous reports, when the size of carbon pores was larger than 1 nm, the normal S_8_ samples would be adsorbed in the inert atmosphere, inevitably giving rise to the formation of dissolved by-products [15]. Interestingly, through the molecules’ simulation of cylinder pore models, it could be noted that the relatively small S molecules (S_2_, S_4_) would react without the excited by-products. Additionally, the relative reactions have been deemed as quasi-solid-state models [16]. On the other hand, relative low capacity could be noted, mainly ascribed to the relative low S-loading and the difficult metal insertion. As is well known, the micro-pores of carbon materials could be divided into four models: cylindrical, slits, conical and bottle neck, which serve important roles in electrochemical properties. According to the previous reports, the pores mainly derive from the in situ formation (like the aggregated cavity structure) and ex situ reaction (such as the etched pores with the assistance of strong alkali) [17]. Compared to the former, the etched pores always displayed the through hole, which hardly brought about the capturing of dissolved polysulfide. Thus, the design strategy of carbon materials can be summed that the suitable thickness was beneficial for the porous formation [18]. That is to say, the through holes always existed in ultra-thin matrix, while the internal pores were hardly created in the thick carbon materials. Thus, the design of carbon materials with suitable thickness were vital for Na-S systems.

Herein, through the in situ polymerizations, the uniform hollow sphere with rich conical-type pores was prepared and then used as hosting matrix for RT Na-S systems. With the assistance of vacuum sealing systems, the as-resulted samples displayed a capacity of 920 mAh g^−1^ after 100 loops. Even at 10 C, the cycling stability could be still prolonged to 7000 cycles. A detailed analysis about the excellent ion-storage abilities are discussed in depth discussed as follows.

## 2. Results

In order to explore the designing advantages of materials, the relevant physical-chemical tests were carried out as shown in Figure 1. The XRD patterns of the as-resulted samples are displayed in Figure 1a. It could be noted that HCS samples displayed amorphous carbon materials. Differing from the graphite with compact structure, the amorphous carbon materials always show abundant micro-porous quality, perhaps bringing about the strong capturing of polysulfide. Interestingly, compared to the peaks (23°) of HCS, the positions of HCS@S-H and CF@S have an obvious moving towards high degree, meanwhile that of HCS@S-P were maintained at similar locations [19]. It could be deduced that, with the assistance of vacuum tube sealing, the gasified sulfur displayed high activities at 650 °C, further inserting into the interlayer accommodations, bringing about the expansion of layer about graphite crystallites [15,20,21]. Meanwhile, the obvious S peaks were not found, further confirming that there is no sulfur on the surface of carbon materials, facilitating their electrochemical properties. Moreover, the relative specific surface area is displayed in Figure 1b, where the pores of CF were cylindrical-type, and those of HCS were slits-type and conical-type. As expected, the through pores could hardly captured polysulfide. Moreover, the as-resulted CF samples displayed high specific surface area (1756 cm^3^ g^−1^) due to the etching of alkali metal ions. Obviously, if the CF were complexed with sulfur at 155 °C, the high loading of sulfur would be obtained. However, at 650 °C, the sulfur mainly existed in porous areas, further demonstrating the advantages of vacuum tube sealing methods and the importance of porous design. Regarding their porous distributions, it could be found that, that of HCS were mainly smaller than 1 nm, mainly resulted from the interspace of in situ long-chain polymerizations. After complexing with sulfur, the micro-pores obviously disappeared, further verifying the successful insertion of sulfur in HCS. Further exploring their detail internal structure, Raman spectra were taken. Among them, it could be noted that D-peaks and G-peaks were found at 1380 cm^−^^1^ and 1580 cm^−^^1^, which were associated with amorphous carbon and graphitized carbon. Interestingly, with the introduction of sulfur, the peaks have an obvious lowering, which demonstrated the reduced vibration strength of hexatomic ring, perhaps resulting from the bonding of C-atoms and S-atoms. In comparison of the intensity of HCS@S-P, the weaker ones of HCS@S-H were noted, demonstrating the formation of C-S bonds at high temperature. Moreover, the particle size of the as-prepared samples are presented in Figure 1f, where the size of HCS@S-H was a little larger than that of HCS, mainly originating from the slight aggregation at secondary calcining process. The rich surficial group served an important role in the controlling of electrochemical properties, and the surficial groups were further investigated in Figure 1g. Herein, the peaks at 620 cm^−^^1^, 1165 cm^−^^1^ and 1210 cm^−^^1^ were noted, indicating the existence of C-S bonds and C-N bonds. Among them, C-N bonds displayed the strong polarity, which would be beneficial for the capturing of polysulfide. Through the analysis of TG curves, it could be noted that the loading of sulfur was about 42%, meeting with the demanding of Na-S systems. Based on the discussion above, the simple mechanism was proposed, where the hollow sphere was successfully prepared. Owing to the complexing of S at high temperature (650 °C), the short-chain S molecule existed in the interlayer accommodation and excited micro-pores with effective geometry confinement. Moreover, the corresponding C-S and C-N bonds existence were further confirmed, boosting the capturing of polysulfide [22,23].

To further explore the surficial traits, XPS spectra of HCS@S-H were investigated, as shown in Figure 2. As expected, the peaks of S, C, O, N and Na elements were noted. As presented in Figure 2b, the peaks of C1s could be divided into five. From the previous reports, it could be noted that the peak located at 284.5 eV was associated with C-C/C=C bonds, mainly resulting from the carbon matrix [24,25,26]. Moreover, the peaks suited at 287.5 eV, 289.1 eV and 290.8 eV were related to C-O, C=O and O-C=O bonds. Moreover, the introduction of hetero-atoms could induce the enhancements of polarity, boosting the capturing of dissolved Na_2_S_n_ intermediates. Therefore, owing to the rich electron pair of O-containing groups, the strong capturing abilities of electrons could bring the improvements of pseudo-capacitive behaviors, finally resulting in positive results in the fast charge/discharge rate abilities. Interestingly, the peak at 286 eV was noted, indicating the formation of C-S bonds, matching well with the results above. In addition, for S 2p core-level XPS, the peaks at 163 eV and 164.9 eV were about S 2p_3/2_ and S 2p_1/2_. Compared to the peaks of traditional C@S samples, they have a slight blue shifting, further indicating the formation of C-S bonds. Further, the peaks at 168.3 eV was assigned to sulfate, perhaps derived from the oxidation of S-elements (about SO_4_^2−^). Interestingly, some Na-S bonds were detected at ~1072 eV, illustrating the formation of Na_2_S, where Na-elements mainly come from raw materials. As expected, the traditional S active material was always limited with the serious expansion, while Na_2_S samples displayed excellent shrinking, which is in favor of their ion-storage abilities. Meanwhile, for O 1s peaks, the peaks at 532.1 eV and 533.4 eV were mainly associated with C=O and C-O bonds, where the peak at 537.1 eV was related with the absorbed oxygen from the atmosphere. In addition, for N 1s peaks, those at 398.6 eV, 400.8 eV and 402.1 eV were about pyridinic-N, pyrrolic-N and graphitic-N, respectively, facilitating the enhancements of conductivity.

Of course, the design of the morphology and internal structure served crucial energy-storage roles. It could be noted that the uniform hollow sphere was prepared without serious aggregation, which could be beneficial for the infiltration of electrolyte and the alleviating of expansion swelling. Through the measurement of Nano-spheres, their size (about 200 nm) could be found. Interestingly, after the calcining and S-complexing, the breakage of hollow-sphere could be noted, perhaps resulting from the shrinking of organic pyrolysis and the vacuum pressure, which could enable the improvements of contacting area between active materials and electrolytes. Besides this, no obvious S materials were noted on the surface of hollow spheres, indicating the successfully complexing of carbon and S materials. Further investigating the shell layers, the thickness about 20 nm could be detected, facilitating the ions diffusions. More significantly, the internal pores serve the important roles in the sulfur-hosting with comparison to the ultra-thin and thin layers. Similar to the discussions, when the layer was ultra-thin, the as-resulted through pores as space confinements hardly limited the shuttling of polysulfide [27,28]. As the layer was too thick, the depth pore could be not effectively filled by S-samples. Further, in the inset images of Figure 3C2, the uniform distribution of S was found, indicating the homogeneous existing of S and further demonstrating of the effective designing of pores. Meanwhile, Figure 3C3 displays the HRTEM images of HCS@S-H samples, where no black dots stated the great complexing manners. In addition, through the analysis of element mapping, the uniform distribution elements could be noted, further disclosing the good design of the morphology of HCS@S-H. Meanwhile, the preparation processes image of HCS and HCS@S were exhibited in Figure 3E, and it could not ignored that the special S-loading method maintained a tight connection with the degree of its cycling-stability.

Driven by the significant structural design, the relative electrochemical properties were carried out. Firstly, it could be found that the designed HCS@S-H samples displayed a high capacity of 920 mAh g^−^^1^ at 0.1 C after 100 cycles, verifying the rich electrochemical activities in Figure 4a. Meanwhile, the capacity of CF@S-H had an obvious fading. From the discussions, the pores of CF@S-H were mainly above 1.0 nm, where the cyclooctasulfur (cyclo-S_8_) usually existed.

In cycling, the dissolved intermediate Na_2_S_n_ (4 ≤ *n* ≤ 8) were shuttling, deteriorated the electrochemical properties. Moreover, owing to the thin layers and the etching of strong alkali, the through pores existed in the carbon matrix, resulting in the loose efficacy of space confinement. Meanwhile, the as-obtained HCS also displayed great cycling abilities, confirming the excellent structural stabilities. Meanwhile, the corresponding differential median curves from 2nd to 100th were displayed in Figure 4d, and the stable shape could be noted, indicating the stable phase change reactions. Based on the previous reports, it could be deduced that the peak at 2.0 V was associated to the reaction (2Na^+^ + S_4_ − 2e^−^→Na_2_S_4_), meanwhile the peak at 1.25 V was in relation to the reactions (Na_2_S_4_ + 2Na^+^ − 2e^−^→2Na_2_S_2_) [29]. As expected, assisted by the vacuum tube sealing methods, the small molecules S_4_ were storage, which could avoid the formation of liquid polysulfide, bringing about the stable electrochemical reactions [30]. When the current density increased up to 1.0 C, the capacity remained at about 740 mAh g^−1^ after 200 cycles in Figure 4b. Interestingly, even at a high rate, HCS@S-H was still stable, indicating their fast redox reaction, which mostly resulted from the high conductive carbon layers. Moreover, it could be found that, after the serious capacity fading, CF@S could hold more capacity than HCS, confirming that the residual S in the micro-pores could contribute to some capacity. Furthermore, in Figure 4e, HCS@S-H displayed the prolonged discharge/charge platforms, disclosing its larger content of S-active materials in comparison of HCS@S-HF. Moreover, at 0.1 C, 0.5 C, 1.0 C, 2.0 C, 5.0 C and 10.0 C, their capacity could still remain at about 980 mAh g^−^^1^, 886 mAh g^−^^1^, 808 mAh g^−^^1^, 751 mAh g^−^^1^, 674 mAh g^−^^1^ and 622 mAh g^−^^1^, respectively, as shown in Figure 4c. Using the capacity at 0.1 C as standard, the residual retention was about 90.4%, 82.4%, 76%, 68% and 63%. Remarkably, with the increasing of current density, many ions/electrons were fast deposited on the surface of electrodes, while the quickening voltage induced the incomplete redox reactions, finally resulting in the lowering of capacity. Compared to the previous work, the as-resulted HCS@S-H samples deliver considerable rate properties. Moreover, with the increasing of current densities, their platforms remained similar, indicating the stable phase changing. From the experience of Dunn’s work, the excellent rate properties mainly came from the rich surficial groups, that was to say, the common electron pair of O-elements enhanced the abilities of electrons gaining/losing [31,32,33,34,35]. More importantly, the uniform distribution of S-elements with high-conductive carbon matrix could induce the enhancements of conductivities. Figure 4g displayed the long-term cycling stabilities; it could be noted that their cycling could be prolonged to 7000 cycles (~310 mAh g^−^^1^), greater than that of other samples from the previous reports as shown in Table 1. Moreover, the stable discharge/charge platforms were noted in the inset of Figure 4c, indicating the fast and stable redox reactions (~180 s). Combining with the discussion above, it could be summarized that the samples displayed rich electrochemical activities and stable structure, further demonstrating the rational architectures.

Obviously, the evolutions of kinetic behaviors were closely related with the improvements of electrochemical properties, which were explored in depth as displayed in Figure 5. For the as-resulted samples, differing from their first cycles, the obvious changing could be noted, mainly coming from the side reactions and the formation of solid electrolyte films. Interestingly, in the curves of CF@S-H and HSC@S-P, the complex oxidation peaks were found from 1.5 V to 2.5 V, corresponding to the reactions from Na_2_S to Na_2_S_2_, Na_2_S_4_ and S_8_ samples, which indicating that some sulfur were only existed on the surface of carbon. As a contrast, relatively simple peaks were noted for HCS@S-H sample, mostly resulting from the rich micro-pores of HCS precursors, further demonstrating that the existing type of sulfur was S_4_. As expected, with cycling, the reduced oxidation peaks were found, where the sulfur had been transformed into liquid polysulfide in the electrolyte. Furthermore, a slight changing was observed in Figure 5C1, indicating excellent cycling stabilities. Further exploring the CV curves at different scan rates, a good linear relationship could be found, confirming the considerable conductivity of carbon materials. Moreover, it could be acknowledged that the ion-diffusion behaviors serve important roles in the electrochemical properties for Na-S systems. From the previous reports, the relative equations have been introduced, about I_p_ = 2.69 × 10^5^ n^2/3^AD^1/2^v^1/2^C [52,53,54,55]. Among them, n is about the transferring number of electrons (2), A is about the contacting area between active materials and electrolyte (0.385 cm^2^), and C is about the Li^+^- concentrations (there are two selections: one is about Li-ions in the electrolyte, another is about the Li-concentration in the lattice). Thus, after fitting, it could be found that the value of diffusion coefficient (Peak-1, Peak-2) was about 2.2 × 10^−^^13^ cm^2^ s^−^^1^ and 1.41 × 10^−^^13^ cm^2^ s^−^^1^ for HCS@S-P, 1.27 × 10^−^^12^ cm^2^ s^−^^1^ and 8.5 × 10^−^^13^ cm^2^ s^−^^1^ for CF@S-H, 5.6 × 10^−^^11^ cm^2^ s^−^^1^ and 9.92 × 10^−^^12^ cm^2^ s^−^^1^ for HCS@S-H, respectively. The results above demonstrate the excellent conductivities of HCS@S, which mainly resulted from two aspects: one is the unique architectures (rich carbon ring), another is the sulfur inserted into the carbon without the surficial residual sulfur. Therefore, as shown in Figure 5H, the diffusion coefficient of the full curves were noted, where the high coefficients were from the main redox reactions, further verifying the uniform distribution of sulfurs. Given the high rate properties, the relative surface-controlling behaviors were explored in detail. Based on the work from Dunn’s group, the equations I = av^b^ were proposed. When b is close to 1, their capacity was determined by pseudo-capacitive behaviors; as b is close to 0.5, their ion-storage abilities were controlled by diffusion behaviors. It could be noted that the b-value of HCS@S-H could reach up to 0.78 and 0.89, whereas those of HCS@S-P and CF@S-H would be up to 0.58/0.63 and 0.71/0.78, respectively, demonstrating the strong surface/near-surface reactions. Moreover, further analyzing their capacitive behaviors (based on the *i* = *k*1*v* + *k*2*v*^1/2^, *i*/*v*
^1/2^ = *k*1*v*^1/2^ + *k*2), the capacitive contribution area was calculated as 75% for HCS@S-H (larger than that of HCS@S-P) at 0.5 mV s^−^^1^. Moreover, with the increasing scan rate, enhanced capacitive behaviors were noted, mainly ascribed to the share electron pairs of O-atoms. Thus, it could be concluded that the excellent electrochemical properties were associated with the unique architecture, which was about the designing of micro-pores and the S-complexing manners.

To further explore the cycling process, the long-term CV curves and SEM image are displayed in Figure 6. Differing from the discussion above, the number of redox peak were obviously reduced. Owing to the relative conductivity of sulfur, the complete redox reactions were hardly triggered, accompanying with the weak side reactions. That was to say, the active materials hardly displayed the ion-storage abilities of themselves, mainly ascribed to the lack of ions-diffusions. Interestingly, compared to the curves of CF@S-H and HCS@S-P, the as-resulted HCS@S-H displayed remarkable peaks, verifying their great electrochemical properties in Figure 6c. Moreover, the similar shapes of HCS@S-H were noted, indicating their excellent cycling stabilities, further demonstrating the advantages of micro-pores. With further increase of the scan rate to 5.0 mV s^−^^1^, the excited peaks were still noted, illustrating their remarkable electrochemical reversibility. Meanwhile, after cycling, their CV curves at different scan rates are displayed in Figure 6e, where the excellent linear relationship is found, indicating their excellent rate properties, matching well with the discussions above. Then, after cycling, SEM images are noted in Figure 6f, and the stable morphology was noted, demonstrating their structural stabilities. Thus, benefitting from the abundant micro-pores and unique S-complexing manners, the active materials successfully inserted into the hosting matrix as S_4_ models without S_8_ residue on their surface, finally resulting in considerable cycling stabilities [56,57].

Certainly, the internal resistances have been regarded as the important roles to evaluate their electrochemical properties, which have been explored in Figure 7. Before cycling, their pristine resistances are displayed in Figure 7a with the relative resistance model. It could be found that the model was composed of the semicircles at high frequency (mainly about the internal contacting resistances) and an oblique line at low frequency (mostly about the ion-diffusion resistance). Regarding this model, R1 is associated with the natural internal resistances, namely, the separators and electrodes. R2 and R3 are the contacting resistances between electrolyte and carbon materials, active materials. Firstly, it could be noted that the internal resistance of CF@S-H was larger than that of other samples, which was attributed to the low conductivities of sulfur. Moreover, the smallest resistances of HCS@S-H were found due to their unique architecture. Based on the equation (D = 0.5 R^2^T^2^/S^2^n^4^F^4^C^2^σ^2^), their diffusion coefficients were further explored. Their slope of w^−^^1/2^ and Z’’ were fitted, and the value of the as-prepared samples were mainly in order of CF@S-H (1090) > HCS@S-P (729) > HCS@S-H (642). Thus, the relative coefficients were calculated about 1.6 × 10^−^^20^ cm^2^ s^−^^1^, 3.01 × 10^−^^20^ cm^2^ s^−^^1^ and 3.89 × 10^−^^20^ cm^2^ s^−^^1^, where their value was different from that about CV curves due to the selection of calculated model. Therefore, their resistances at different cycles were explored at different cycles (10th, 50th, 100th). With cycling, the reduced resistance were detected, which connected with HCS@S-H, mainly attributed to the incomplete redox reactions (that is to say, CE < 100%), where the residual Na-ions would improve the conductivity of electrode materials. Meanwhile, the interesting increase of CF@S-H was noted due to the serious side reactions. Meanwhile, in Figure 7e,f, their corresponding diffusion coefficients were further investigated. As expected, owing to the effective space confinements, the relative stable diffusion coefficients were noted for the HCS@S-H samples.

Triggered by the interesting phase change of S_4_ samples, in situ EIS was performed at different discharged/charged states. Herein, based on their phase changing process, the discharged state was selected, containing 2.99 V, 2.51 V, 2.01 V, 1.56 V, 1.42 V, 0.89 V and 0.56 V, while the charged states were chosen as 0.01 V, 1.54 V, 1.74 V, 2.13 V and 2.55 V. It could be concluded that, with discharging, the interfacial resistances were further increased, mainly ascribed to the formation of weak inductive Na_2_S samples. Moreover, through the charged process, similar plots were noted in Figure 8c, illustrating the stable electrochemical properties and further demonstrating their stable phase change process. Further exploring their diffusion coefficients, the dates of diffusion coefficients of HCS@S-H were displayed in Table 2 and the relative slope curves were obtained and presented in Figure 8e, and which were inversely proportional to the value of diffusion coefficients. Moreover, the weak inductive capability of Na_2_S samples and the occupying of pore spaces would hinder their ion moving behaviors, finally resulting in the lowering of diffusion behaviors. However, with the charge process, diffusion coefficients recovered to the pristine value. Herein, it could be noted that, benefitting from the excellent space confinements, S_4_-type active materials were successfully controlled with stable electrochemical properties.

## 3. Materials and Methods

### 3.1. Materials

Chemicals: Calcium citrate, 2,4-Dihydroxybenzonic acid (DA), Hex methylene tetramine (HMT), Sodium oleate (SO), and Elemental sulfur were purchased from Macklin Reagents Ltd. P-123 was purchased from Sigma-Aldrich Reagents Ltd. (St. Louis, MI, USA). All chemical reagents were used in the preparation of the samples without further purification.

Synthesis of HCS and HCS@S-H: We mixed the two types of solutions (0.616 g DA and 0.232 g HMT were dissolved separately in 120 mL deionized water, and 0.146 g SO and 0.087 g P-123 were dissolved in 40 mL deionized water under stirring to form a transparent solution) to obtain a light blue emulsion. Then, the mixture was transferred into a 200 mL Teflon-lined hydrothermal autoclave, followed by thermal treatment where the temperature slowly increased to 160 °C with a heating rate of 5 °C min^−^^1^ and the sample was maintained at this temperature for 8 h, followed by cooling to room temperature. Black HCS precursor was separated after centrifuging the slurry solution at 15,000 rpm, then washed with deionized water 2–3 times. Under vacuum, the precipitate was dried at 70 °C for 12 h, and then the as-prepared powder sample was calcined under Argon atmosphere at 650 °C for 3 h to obtain HCS. HCS@S-H was outputted by calculating the ground mixture (HCS: S = 1:1 wt%) in a vacuum-sealed glass tube at the tubular furnace under 500 °C for 2 h with a heating rate of 5 °C min^−^^1^.

Synthesis of HCS@S-P: HCS@S-P was yielded by calculating the ground mixture (HCS: S = 1:1 wt%) under argon atmosphere at 155 °C for 24 h with a heating rate of 5 °C min^−^^1^.

Synthesis of CF and CF@S-H: A certain amount of Calcium citrate was incinerated at 650 °C under the argon atmosphere for 3 h with a heating rate of 5 °C min^−^^1^ in the tube furnace to produce the CF. The same progress prepared CF@S-H as HCS@S-H.

### 3.2. Material Characterizations

The morphologies of the samples were investigated by SEM (FEI Quanta FEG 450) and TEM (JEOL 2011, 200 keV). XRD patterns were collected by powder XRD (Bruker, Billerica, MA, USA, D8 ADVANCE) with Cu Kα radiation at a scan rate of 5 °C min^−^^1^. XPS (Escalab Xi^+^, Thermo Fisher Inc., Shanghai, China.) measurements and FT-IR (Bruker vertex FT-IR spectrophotometer) was carried out using Al Kα radiation to characterize the valence states of the elements and functional groups. Raman spectra were collected on LabRAMHR800 with a 532 nm laser. Thermogravimetric analysis (TG, air atmosphere, the heating rate of 10 °C min^−^^1^, test temperature range from 25 °C to 1000 °C) and Particle size analysis (Hydro 2000 MU (A)) were tested. To gain detailed aperture and specific surface area data, automatic specific surface and porosity analyses were carried out on BELSORP-MINIⅡ (N_2_, degassing temperature 280℃, degassing time 3 h).

### 3.3. Electrochemical Measurements

To fabricate Na-S batteries, 70 wt% carbon/sulfur composites, 15 wt% acetylene black and 15 wt% carboxymethyl cellulose solution were blended in a small glass vial. The formed slurry was coated on a copper foil by applying the doctor blading, which was further vacuum dried at 75 °C overnight. CR2016 coin cells were assembled in an Ar-filled glove box, with Na metal foil as counter electrodes and glass fiber (GF/D, Whatman, Maidstone, UK) as the separator. The electrolyte was prepared by dissolving 1 M NaClO_4_ in EC: PC = 1:1 Vol% with 5 wt% FEC. Blue battery test system (CT2011) was used to test constant current charge-discharge and rate performance at different current densities in the voltage range of 0.01–3.0 V. Cyclic voltammetry (CV) was obtained at different scanning rates on the Multi Autolab 204 electrochemical workstation. Electrochemical impedance spectroscopy (EIS) was performed in the frequency range of 0.01 to 100 kHz, and all electrochemical tests were performed at room temperature. The loading weight of the active material was up to 0.7 mg cm^−^^2^.

## 4. Conclusions

In summary, it could be noted that the uniform hollow spheres were successfully prepared through hydrothermal methods. Assisted by the vacuum sealing techniques, active materials were inserted into the micro-pores of hosting materials as S_4_-type. Compared to other samples, the as-resulted samples displayed suitable thickness of carbon layers, facilitating the inserting behaviors towards in-depth spaces. Moreover, the C-S and C-N bonds were noted, accompanying the improvements of conductivity. Further, the abundant O-containing groups were noted, bringing about the enhancements of ion/electrons capturing abilities. Used as electrodes materials for Na-S systems, the excellent cycling stabilities could be maintained about 920 mAh g^−^^1^ after 100 cycles at 0.1 C. When the current density increased up to 1.0 C, the capacity remained at about 740 mAh g^−^^1^ after 100 cycles. Even at 10 C, their capacity could remain at 310 mAh g^−^^1^ after 7000 cycles. Through the analysis, it could be noted that the residual S_8_ without the physical trapping of micro-pores would induce the fast “shuttling effect” of polysulfide. Moreover, after calculations, their diffusion coefficient was about 3.3 × 10^−12^ cm^2^ s^−^^1^, much larger than that of other samples, mainly ascribed to the effective S-complexing process. Meanwhile, supported by the detailed kinetic exploring, the enhanced rate properties were determined by pseudo-capacitive behaviors. Therefore, the stable phase changing process was further confirmed through in situ resistances. Given this, the work is expected to shed light on the design of hosting materials and complexing manners towards great energy-storage properties.

## Figures and Tables

**Figure 1 molecules-27-05880-f001:**
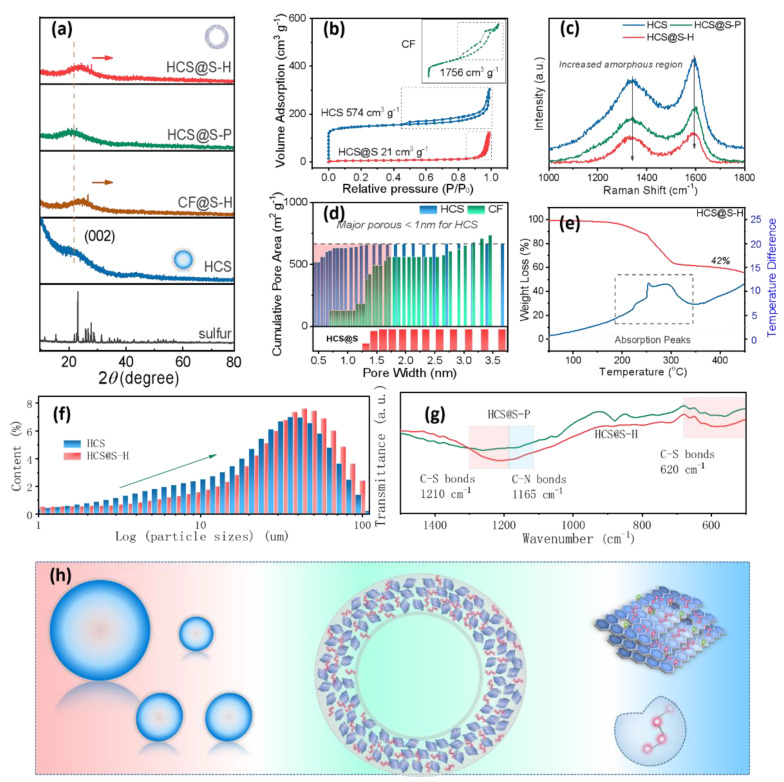
The physical-chemical properties of the as-resulted samples: XRD (**a**), specific surface area (**b**), Raman (**c**), porous distribution (**d**), TG curves (**e**), particles size distribution (**f**), FTIR (**g**) and the formation mechanism (**h**).

**Figure 2 molecules-27-05880-f002:**
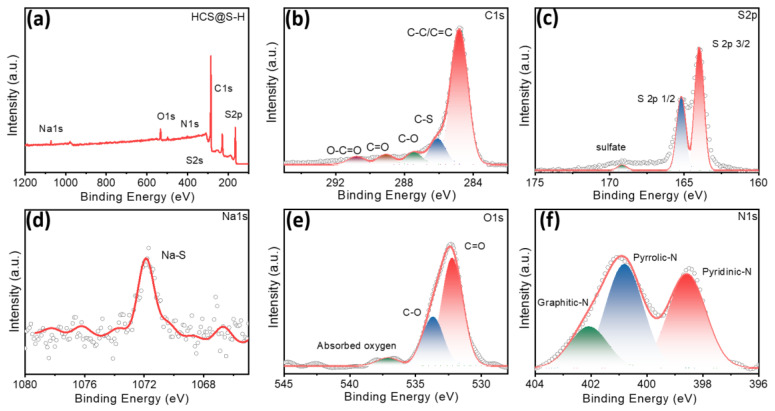
XPS spectra of HCS@S-H: Full spectra (**a**), C1s (**b**), S2p (**c**), Na1s (**d**), O1s (**e**) and N1s (**f**).

**Figure 3 molecules-27-05880-f003:**
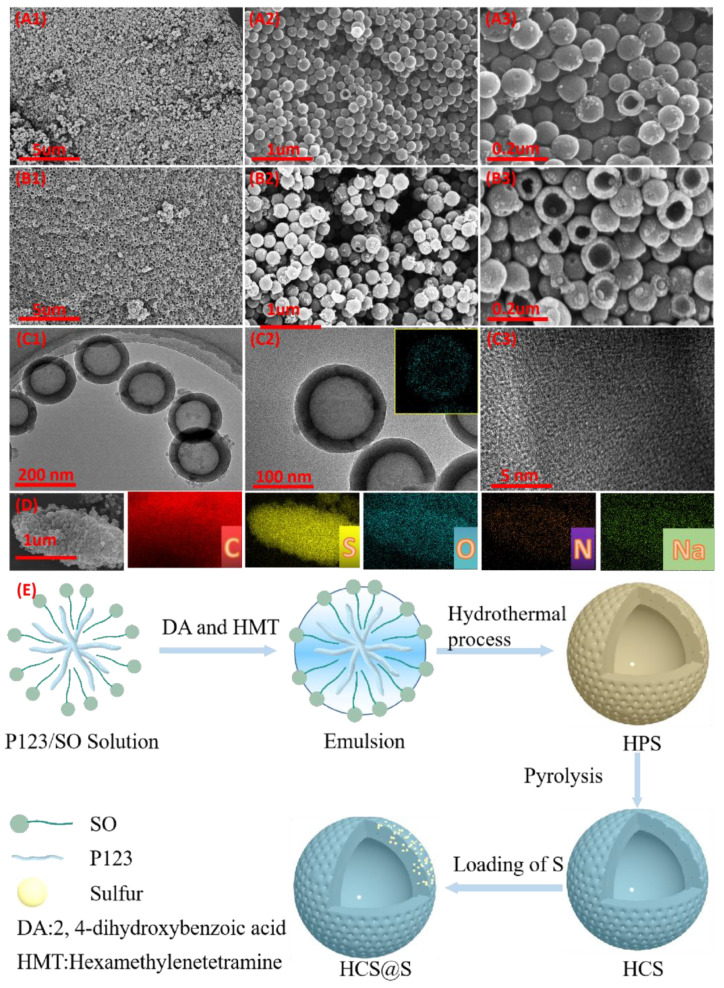
The exploration of their morphology: SEM images of HCS (**A**) and HCS@S-H (**B**), TEM images and mapping images of HCS@S-H (**C**,**D**) and the preparations process of HCS and HCS@S (**E**).

**Figure 4 molecules-27-05880-f004:**
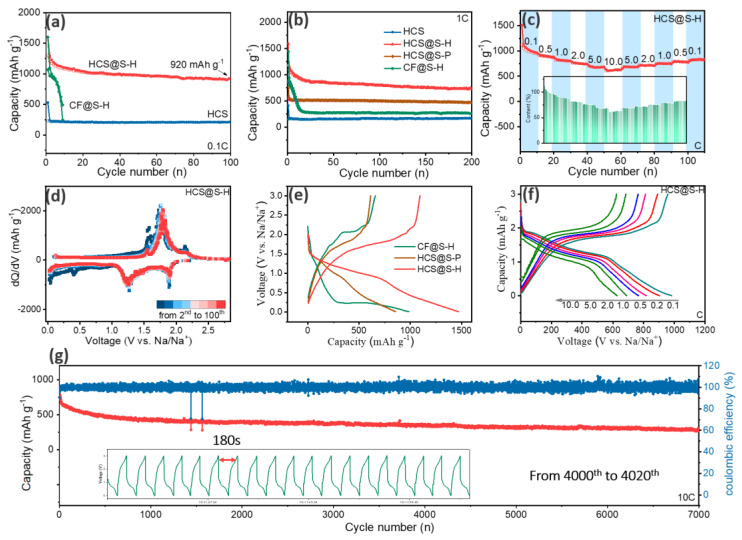
The investigation of electrochemical properties: Cycling stabilities at 0.1 C (**a**) and 1.0 C (**b**), rate properties of HCS@S-H (**c**), differential median curves at different cycles (**d**), the 10th discharge/charge platforms (**e**), the platforms of HCS@S-H at different cycles (**f**) and the long-term cycling stabilities at 10.0 C (**g**).

**Figure 5 molecules-27-05880-f005:**
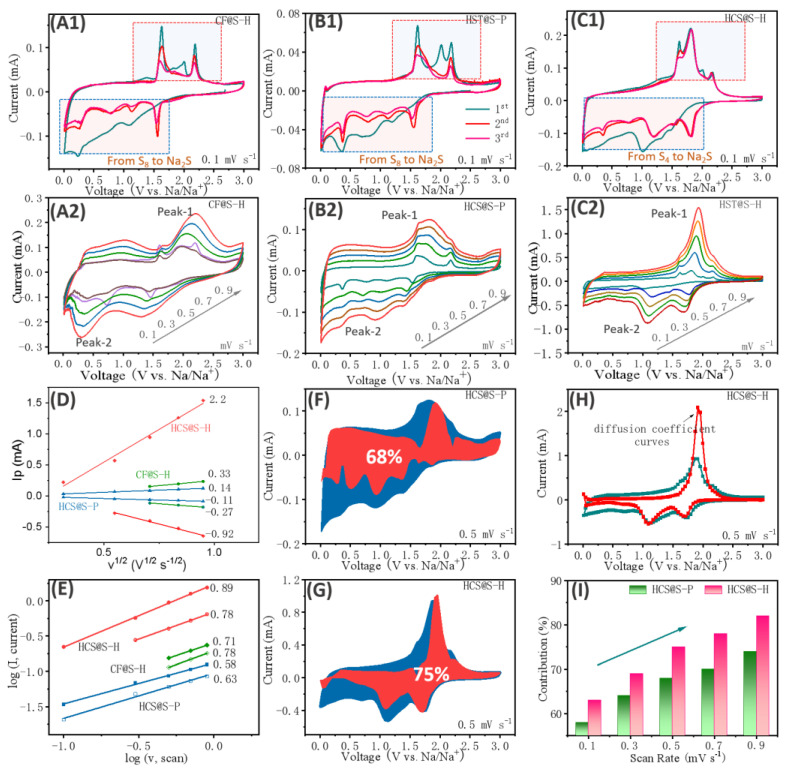
The analysis of kinetic behaviors: CV curves at 0.1 mV s^-1^and different scan rates for CF@S-H (**A1**,**A2**), HCS@S-P (**B**), HCS@S-H (**C**), the linear relationship of peak-current and v^1/2^ (**D**), log (i) and long (v) (**E**), the pseudo-capacitive contributions area at 0.5 mV s^-1^ for HCS@S-P (**F**) and HCS@S-H (**G**), the diffusion coefficients about the full curves about HCS@S-H (**H**) and the surface-controlling capacity contribution at different scan rates (**I**).

**Figure 6 molecules-27-05880-f006:**
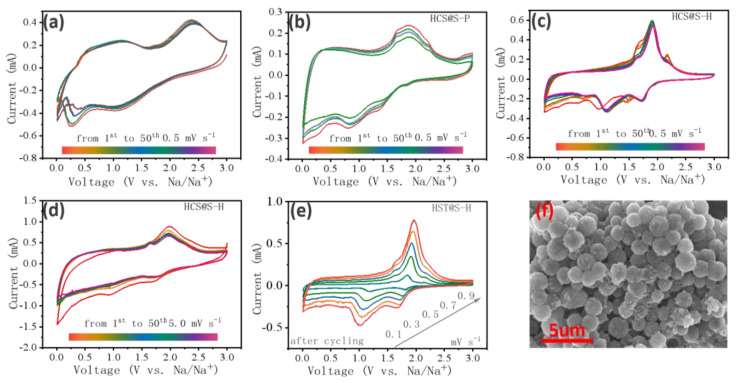
The long-term cycles from 1st to 50th 0.5 mV s^-1^ for CF@S-H (**a**), HCS@S-P (**b**) and HCS@S-H (**c**), at 5.0 mV s^-1^ for HCS@S-H (**d**), CV curves after cycling (**e**) and the morphology after cycling for HCS@S-H, The Mapping after 20 cycling loops of HCS@S-H (**f**).

**Figure 7 molecules-27-05880-f007:**
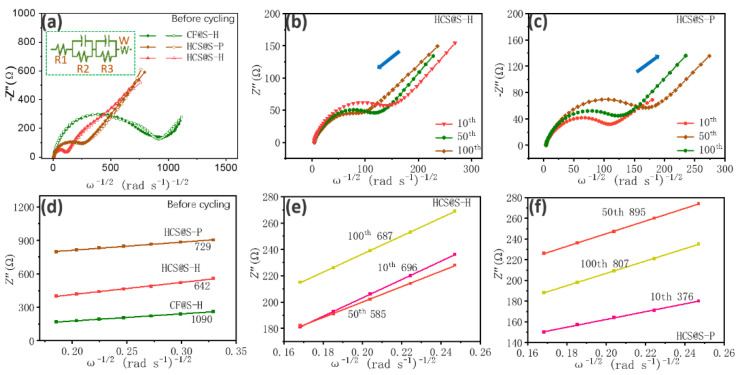
Nyquist plots of the as-resulted samples. Nyquist plots and fitting lines before cycling (**a**), Nyquist plots at different cycles for HCS@S-H (**b**), CF@S-H (**c**), their linear relationship between w^−1/2^ and Z’’ for the as-resulted samples before cycling (**d**), at different cycles for HCS@S-H (**e**) and CF@S-H (**f**).

**Figure 8 molecules-27-05880-f008:**
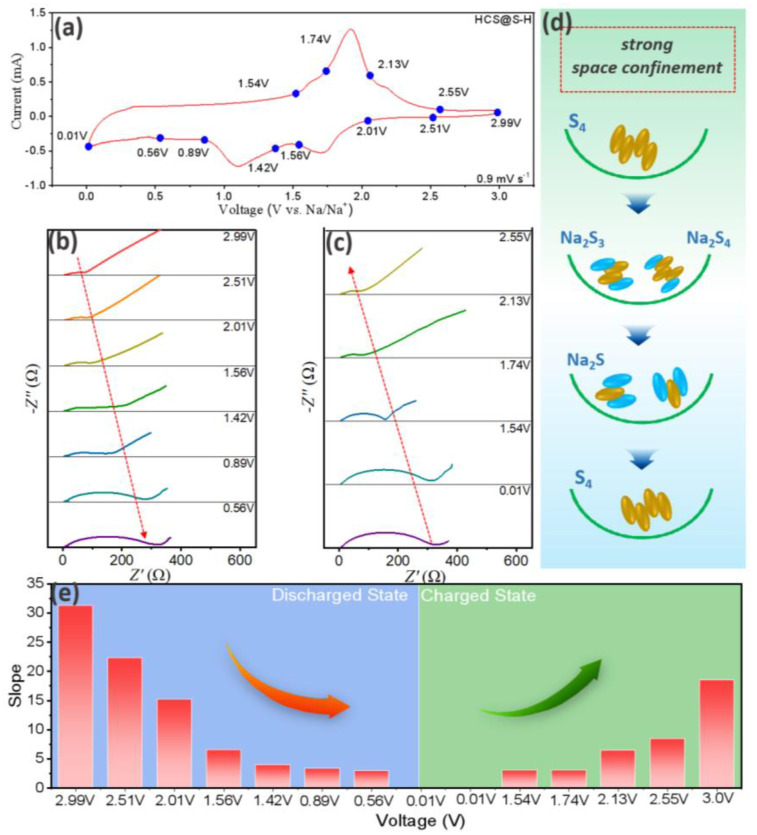
The exploration of resistances during charged/discharged process, CV curves at 0.9 mV s^−1^ (**a**), Nyquist plots at discharged state (**b**), at charged state (**c**), the relative energy-storage behaviors (**d**) and the slope at different discharged/charged states (**e**).

**Table 1 molecules-27-05880-t001:** The electrochemical properties for the previous reports.

Materials	Manners	CE (%)	Current Density (A g^−1^)	Cycles (n)	Capacity (mA h g^−1^)	Ref.
Ti_3_C_2_Tx MXene Microspheres	spray drying and annealing	70.28	3.3	500	450.1	Ref [36]
carbon fiber@S	Solution method	70.6	0.167	400	997	Ref [37]
Sulfurized Polyacrylonitrile	Electrospinning	90	0.167	100	1405	Ref [38]
Macro-microporous carbon@S	calcination	40.24	1.67	510	330	Ref [39]
MoS_2−x_ nanosheets and hollow carbon spheres@S	Hydrothermal method	85.2	2.0	100	415.7	Ref [40]
Yolk@Shell Nitrogen-Doped Carbon@S	carbonate precipitation method	75	0.5	330	636.1	Ref [41]
S@Ni/Co-C	Hydrothermal method	86.8	0.5	200	813.5	Ref [42]
FeNi_3_@HC@S	Solution method		2.0	500	591	Ref [43]
S/MnS/NCS	one-pot chemical wet method	99	0.2	800	774.2	Ref [44]
CoS/NC@S	Hydrothermal method	50.72	0.1	100	488	Ref [45]
Co@PCNFs/S	Solution method		0.84	600	398	Ref [46]
S/TiN-TiO_2_@MCCFS	Electrospinning	48.92	0.1	100	640	Ref [47]
yolk-shell Fe_2_N@nitrogen doped carbon	Hydrothermal method	66.8	1	200	603	Ref [48]
S@Co-CNT/NPS	Solution method		0.5	1000	351.8	Ref [49]
Microporous carbon spheres@S	Solution method	63.4	0.84	430	718	Ref [50]
Mmpcs@S	chemical reaction method	61.0	2.0	2000	420	Ref [51]
this work	Hydrothermal method	89	8.5	7000	310	

**Table 2 molecules-27-05880-t002:** The value of diffusion coefficients at different coefficients about HCS@S-H.

Discharge State (V)	2.99	2.51	2.01	1.56	1.42	0.89	0.56
**Diffusion coefficients (** **10^−14^ cm^2^ s^−1^)**	0.95	1.86	3.99	22.1	58	82	104
**Charge State (V)**	0.01	1.54	1.74	2.13	2.55		
**Diffusion coefficients (** **10^−14^ cm^2^ s^−1^)**	100.3	97.6	22.3	13.7	2.7		

## Data Availability

Not applicable.

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
