# Peer review of "Designing Hollow Carbon Sphere with Hierarchal Porous for Na-S Systems with Ultra-Long Cycling Stabilities"

_molecules, 2022, doi:10.3390/molecules27185880_

Round 1

Reviewer 1 Report

This work focused on the use of S complex in the pores of hollow carbon spheres as a negative electrode for sodium-sulfur batteries. Although powerful characterization techniques such as XRD, Raman and X-ray photoelectron spectroscopy are used, the production and layout of the picture is also relatively beautiful, this work still has many unclear points in the manuscript. Therefore, I disagree to publish this manuscript in Molecules. The comments are as follows:

1.    As shown in Figure 1(b), the specific surface area of CF is much larger than that of the other two materials, and a large specific surface area can shorten the ion transport path, reduce the surface current density, and improve the kinetic performance of the battery. Why is the electrochemical performance of CF inferior to the other two materials?

2.    In line 117, a protruded peak appears in Figure 1(e), but there is no obvious peak in the figure, and there is a lot of noise. Is this description too far-fetched?

3.    The schematic diagram in Figure 1(i) is too simplistic to make it obvious to the reader without a corresponding process.

4.    This work has repeatedly mentioned the complex of S into the micropores of the carbon hollow sphere. SEM images should represent the micropores of CF and HCS, and mapping images should show the distribution of elements of a carbon hollow sphere, instead of placing a random picture without knowing which area it is.

5.    In Figure 7(b), the impedance decreases gradually as the cycle progresses, but why does the fitted data in Figure 7(e) show a minimum at 50 cycles? Is the data wrong?

6.    In addition, there are still many details in the manuscript that need to be revised. For example, ‘porous distribution (c)’ in Figure 1 should be changed to ‘porous distribution (d)’; The sweep speeds of 0.1 and 0.3 mV s-1 are missing in Figure 5(A2); The font of the coordinates in Figure 6(a) is different from the others; In Figure 8, the abscissa of b and c overlaps with e.

Author Response

Response to Reviewer 1 Comments

Point 1: As shown in Figure 1(b), the specific surface area of CF is much larger than that of the other two materials, and a large specific surface area can shorten the ion transport path, reduce the surface current density, and improve the kinetic performance of the battery. Why is the electrochemical performance of CF inferior to the other two materials?

Response 1: Thank you very much! It was known that, the larger specific surface area was helpful for the transportation of ion or electronic. However, for CF, the relative larger area was resulted by the wide pore width, which hardly restrained the sulfur, accompanying with shuttle effect and capacity fading. At the same time, there were number of perforations for CF, leading to an artificial high specific surface area and it was unable to became the container of active substance, finally resulting in the inferior energy storage performance. Meanwhile, some sulfur on the surface of CF were diffucult to deliver the capacity, but seriously deteriorating electrodes.

Point 2: In line 117, a protruded peak appears in Figure 1(e), but there is no obvious peak in the figure, and there is a lot of noise. Is this description too far-fetched?

Response 2: Yes, the relative figure have been revised as suggested.

Point 3: The schematic diagram in Figure 1(i) is too simplistic to make it obvious to the reader without a corresponding process.

Response 3: The details of synthesis processes and loading process have been added in the revised revisions. 

Point 4: This work has repeatedly mentioned the complex of S into the micropores of the carbon hollow sphere. SEM images should represent the micropores of CF and HCS, and mapping images should show the distribution of elements of a carbon hollow sphere, instead of placing a random picture without knowing which area it is.

Response 4: Thanks for your efforts to improve the paper. The EDS mapping images have been revised, clearly showing the element distribution.

Point 5: In Figure 7(b), the impedance decreases gradually as the cycle progresses, but why does the fitted data in Figure 7(e) show a minimum at 50 cycles? Is the data wrong?

Response 5: Thanks, the relative figure have been revised as suggested. Moreover, owing to the precision of the fitting data, they were always regarded as the evaluated tendency. Herein, the data were used to distinguish the stability of the electrochemcial process. Through the comparsion of Fig.7e and Fig.7f, the relative stable resistances of HCS@S-H could be noted, further demonstating their merits of structural designing.   

Point 6:  In addition, there are still many details in the manuscript that need to be revised. For example, ‘porous distribution (c)’ in Figure 1 should be changed to ‘porous distribution (d)’; The sweep speeds of 0.1 and 0.3 mV s-1 are missing in Figure 5(A2); The font of the coordinates in Figure 6(a) is different from the others; In Figure 8, the abscissa of b and c overlaps with e.

Response 6: Thank you for your useful comments and kind suggestions to further improve our manuscript. The paper fo the whole paper have been carefully revised as suggested.

Reviewer 2 Report

 Wang et al reported 2,4-Dihydroxybenzonic acid derived porous hollow carbon sphere for Na-S systems with ultra-long cycling stability. The work is interesting and the presentation of findings is also impressive. The MS can be accepted for publication after minor revision. 

1. Why did the author choose different pyrolysis temperatures for the preparation of HCS@S-H, CF@S-H and HCS@S-P?

2. The author includes the mechanism of the formation of sphere-shaped structure?

3. Fig. 3, Author should provide high resolution EDS mapping images.  Because it is very difficult to observe the presence of elements in the images. 

Author Response

Response to Reviewer 2 Comments

Point 1: Why did the author choose different pyrolysis temperatures for the preparation of HCS@S-H, CF@S-H and HCS@S-P?

Response 1: In the paper, the CF and HCS have been used to load ative materials as precursors. Herein, compared to the traditional loading manners, the new-type vacuum sealing method has been used to improve the meirts of structural designing.

Point 2: The author includes the mechanism of the formation of sphere-shaped structure?

Response 2: The image of mechanism of the formation of sphere-shaped structure was sucessfully provided.

Point 3: Author should provide high resolution EDS mapping images.  Because it is very difficult to observe the presence of elements in the images. 

Response 3: The EDS mapping images have been revised as suggested, showing the clear element distribution.

Round 2

Reviewer 1 Report

It is suggested more references should be reviewed and a comprehensive discussion should be given. I agree to accept this manuscript for publication after minor revision.